# Usefulness of Machine Learning-Based Gut Microbiome Analysis for Identifying Patients with Irritable Bowels Syndrome

**DOI:** 10.3390/jcm9082403

**Published:** 2020-07-27

**Authors:** Hirokazu Fukui, Akifumi Nishida, Satoshi Matsuda, Fumitaka Kira, Satoshi Watanabe, Minoru Kuriyama, Kazuhiko Kawakami, Yoshiko Aikawa, Noritaka Oda, Kenichiro Arai, Atsushi Matsunaga, Masahiko Nonaka, Katsuhiko Nakai, Wahei Shinmura, Masao Matsumoto, Shinji Morishita, Aya K. Takeda, Hiroto Miwa

**Affiliations:** 1Division of Gastroenterology and Hepatology, Department of Internal Medicine, Hyogo College of Medicine, 1-1, Mukogawa, Nishinomiya 663-8501, Japan; hfukui@hyo-med.ac.jp (H.F.); miwahgi@hyo-med.ac.jp (H.M.); 2Cykinso Inc., 1-36-1, Yoyogi, Shinjuku, Tokyo 151-0053 Japan; nishida.a.aa@aoni.waseda.jp (A.N.); ki9163724121@yahoo.co.jp (F.K.); s.watanabe@cykinso.co.jp (S.W.); kuriyama@cykinso.co.jp (M.K.); 3Department of Electrical Engineering and Bioscience, Waseda University, 1-104, Totsuka, Shinjuku, Tokyo 169-8050, Japan; 4School of Computing, Tokyo Institute of Technology, 2-12-1, Okayama, Meguro, Tokyo 152-8550, Japan; 5Colo-proctological Institute, Matsuda Hospital, 753, Irino-cho, Nishi-ku, Hamamatsu, Shizuoka 432-8061, Japan; tsudamatsu176cm74kg@yahoo.co.jp (S.M.); gaku@matsuda-hp.or.jp (K.K.); aiko-a@matsuda-hp.or.jp (Y.A.); oda@matsuda-hp.or.jp (N.O.); dgsdg272@yahoo.co.jp (K.A.); j1matsunaga@yahoo.co.jp (A.M.); nonaka@matsuda-hp.or.jp (M.N.); nakai@matsuda-hp.or.jp (K.N.); 6Department of Gastroenterology, JCHO Tokyo Shinjuku Medical Center, 5-1, Tsukudo, Shinjuku, Tokyo 162-8543, Japan; shinmura62@gmail.com (W.S.); m.masao1210@outlook.com (M.M.); morishita-shinji@shinjuku.jcho.go.jp (S.M.)

**Keywords:** IBS, gut microbiome, short-chain fatty acids, machine learning

## Abstract

Irritable bowel syndrome (IBS) is diagnosed by subjective clinical symptoms. We aimed to establish an objective IBS prediction model based on gut microbiome analyses employing machine learning. We collected fecal samples and clinical data from 85 adult patients who met the Rome III criteria for IBS, as well as from 26 healthy controls. The fecal gut microbiome profiles were analyzed by 16S ribosomal RNA sequencing, and the determination of short-chain fatty acids was performed by gas chromatography–mass spectrometry. The IBS prediction model based on gut microbiome data after machine learning was validated for its consistency for clinical diagnosis. The fecal microbiome alpha-diversity indices were significantly smaller in the IBS group than in the healthy controls. The amount of propionic acid and the difference between butyric acid and valerate were significantly higher in the IBS group than in the healthy controls (*p* < 0.05). Using LASSO logistic regression, we extracted a featured group of bacteria to distinguish IBS patients from healthy controls. Using the data for these featured bacteria, we established a prediction model for identifying IBS patients by machine learning (sensitivity >80%; specificity >90%). Gut microbiome analysis using machine learning is useful for identifying patients with IBS.

## 1. Introduction

Irritable bowel syndrome (IBS) is currently accepted to be a functional gastrointestinal disorder characterized by symptoms such as abdominal pain or discomfort, bloating, and stool irregularities without any structural or organic lesions [1]. Many factors (visceral sensitivity, bowel motility, mucosal immunity, psychological stress, etc.) are involved in the pathophysiology of IBS [2], making it difficult to clarify the pathophysiological mechanism, diagnosis, and treatments for IBS. With regard to diagnosis, although some objective molecular markers based on blood, stool, and intestinal tissue sampling have been proposed, no valid biomarkers of IBS have yet been established [3]. In this context, the Rome Foundation have developed symptom-based criteria for diagnosing and distinguishing the clinical types of IBS, and the Rome Criteria are now used as a global standard.

The Rome III criteria, revised into Rome IV in 2016 [4], have been used for the diagnosis of IBS since 2006, and numerous data based on the Rome III criteria have been accumulated over the last decade. Ford et al. reported that the sensitivity and specificity of the Rome III criteria for the diagnosis of IBS are 68.8% and 79.5%, respectively [5,6]. This suggests that the Rome III criteria have room for further improvement, and, in fact, it is possible to categorize inflammatory bowel disease (IBD) or celiac disease into IBS on the basis of the Rome III criteria [7,8]. Thus, there is still a need for objective biomarkers with improved diagnostic accuracy for IBS, which can help identify individuals who will develop IBS in the future. Recent studies have strongly suggested that the gut microbiome may play a pivotal role in the pathophysiology of IBS [9,10]. In this context, the possibility that the intestinal microbiota signature might be a candidate biomarker for evaluating the severity of IBS symptoms has been suggested by a European group [11]. However, further detailed studies are needed to clarify whether the microbiota signature could be useful for diagnosis, group typing, and evaluation of both clinical severity and response to therapy. On the other hand, it is well known that the gut microbiota profile differs among human racial groups [12]. Therefore, in the present study, we investigated the gut microbiota profile and associated short-chain fatty acids in Japanese IBS patients and healthy controls, its relationship to clinical data and molecular samples, and its possible usefulness as a biomarker in clinical subsets of IBS.

## 2. Materials and Methods

### 2.1. Study Design and Participants

Eighty-five adult patients, aged 20–65 years who fulfilled the Rome III criteria for IBS were recruited prospectively at secondary/tertiary care outpatient clinics (Matsuda Hospital, JCHO Tokyo Shinjuku Medical Center, and Hyogo College of Medicine in Japan) between February 2017 and February 2018. A healthy control group of 26 individuals was also recruited by advertisement and checked by interview and a questionnaire to exclude any chronic diseases and or current gastrointestinal symptoms. All subjects provided written informed consent to participate after receiving verbal and written information about the study. All of the procedures complied with the principles of the Declaration of Helsinki and were approved by the Ethical Review Board at Matsuda Hospital (IRB No. H29-2), JCHO Tokyo Shinjuku Medical Center (IRB No. 2016-04) and Hyogo College of Medicine (IRB No. 2700).

Demographic information and body mass index were collected from all subjects. They were also asked to complete a questionnaire designed to obtain information about medical and medication history. For IBS patients, the Bristol stool form scale score and the characteristics and frequency of gastrointestinal symptoms were recorded [13]. Classification into IBS subtypes according to the Rome III criteria was performed based on the Bristol Stool Form scale characteristics: IBS with constipation (IBS-C), IBS with diarrhea (IBS-D), mixed IBS (IBS-M), or unsubtyped IBS (IBS-U) [1]. Exclusion criteria for all subjects included (i) use of antibiotics or antacids within one month before inclusion, (ii) having a major psychiatric disorder or use of psychotropic medication within one month before inclusion, and (iii) habitual use of tobacco or alcohol.

### 2.2. Fecal Sampling, DNA Extraction, and Sequencing

Fecal samples were collected using a brush-type collection kit containing a guanidine thiocyanate solution (Techno Suruga Laboratory, Shizuoka, Japan) and stored at 4 °C. DNA was extracted from fecal samples using an automated DNA extraction machine (GENE PREP STAR PI-480, Kurabo Industries Ltd., Osaka, Japan) according to the manufacturer’s standard protocol. The 16S ribosomal RNA (rRNA) regions (V1–V2) were amplified using a forward primer (16S_27Fmod: TCG TCG GCA GCG TCA GAT GTG TAT AAG AGA CAG AGR GTT TGA TYM TGG CTC AG) and reverse primer (16S_338R: GTC TCG TGG GCT CGG AGA TGT GTA TAA GAG ACA GTG CTG CCT CCC GTA GGA GT) with KAPA HiFi Hot Start Ready Mix (Kapa Biosystems, Wilmington, MA, USA). To sequence 16S amplicons by the Illumina MiSeq platform, dual index adapters were attached using the Nextera XT Index kit (Illumina, San Diego, CA, USA). Each library was diluted to 5 ng/µL, and equal volumes were mixed to 4 nM. The DNA concentration of the mixed libraries was quantified by qPCR with KAPA SYBR FAST qPCR Master mix (KK4601, KAPA Biosystems, Wilmington, MA, USA) using primer 1 (AAT GAT ACG GCG ACC ACC) and primer 2 (CAA GCA GAA GAC GGC ATA CGA). The library preparations were carried out according to the 16S library preparation protocol of Illumina (Illumina, San Diego, CA, USA). Libraries were sequenced using the MiSeq Reagent Kit v2 (500 Cycles) for 250-bp pair-ends (Illumina, San Diego, CA, USA). Sequence files are available from the NCBI Sequence Read Archive [14].

### 2.3. Taxonomy Assignment Based on the 16S rRNA Gene Sequence

The paired-end reads of partial 16S rRNA gene sequences were clustered by 97% nucleotide identity, and then assigned taxonomic information using the Greengenes database (v13.8) [15] through the QIIME pipeline (v1.8.0) [16]. The steps for data processing and assignment based on the QIIME pipeline were as follows: (i) joining paired-end reads; (ii) quality filtering with an accuracy of Q30 (>99.9%) and a read length of >300 bp (the number of reads per sample before and after quality filtering is listed in Appendix A); (iii) random extracting of 10,000 reads per sample for subsequent analysis; (iv) clustering of operational taxonomic units (OTUs) with 97% identity by UCLUST (v1.2.22q) [17] (all the relative abundance values for each OTU and sample are listed in Appendix A); (v) assigning of taxonomic information to each OTU using the Ribosomal Database Project (RDP) classifier [18] with the full-length 16S gene data of Greengenes (v13.8) to determine the identity and composition of the bacterial genera.

### 2.4. Analysis of Bacterial Diversity

Microbiota diversity was assessed by Shannon index, PD (phylogenetic diversity) whole tree, and observed OTUs based on 97% nucleotide sequence identity. These values were calculated by QIIME [16] with a depth of 10,000. Then, *p*-values were calculated by Welch’s test for testing group differences in diversity between the IBS patients and healthy controls. All distances among IBS patients and healthy controls were assessed by unweighted UniFrac distance by QIIME. Principal coordinate analysis (PCoA) was used to show the unweighted UniFrac distance between IBS patients and healthy controls in a low-dimensional space by cmdscale in the R statistical platform, version 3.4.3 [19]. Hierarchical clustering of unweighted UniFrac distance using Ward’s method was performed to visualize the relationship between IBS patients and healthy controls using the Python clustering package (Scipy v1.2.1) [20]. All links connecting nodes closer than 2 Euclidean distances were assigned the same color.

### 2.5. Measurement of Fecal Short-Chain Fatty Acids

Fecal samples were collected from all participants. The fecal samples obtained for measurement of short-chain fatty acids (SCFAs) were immediately frozen at −30 °C and stored at −80 °C until measurement. Fecal SCFAs were measured using a modified protocol described previously [21]. In brief, the SCFA-containing ether layers were collected and pooled for gas chromatography–mass spectrometry (GC/MS) analysis using GCMS-QP2010 Ultra (Shimadzu, Kyoto, Japan). The concentration of each SCFA was determined as µmol/g using external standard calibration over an appropriate concentration range. A *p*-value by Welch’s test of <5% was considered to be significant.

### 2.6. Group Differences in Taxonomic Abundance

To reveal associations between taxonomic abundance and IBS status, we tested group differences of genus-level relative abundances using Welch’s test. The centered log-ratio transformed values were used as inputs for these univariate analyses to manage 0 count values. Analysis was confined to taxa with a prevalence greater than 10% and a maximum proportion (relative abundance) greater than 0.005. A *p* value of less than 5% was considered to be significant.

### 2.7. Prediction Model for IBS and Statistical Analyses of IBS Biomarkers

To establish a methodology for identifying IBS patients based on fecal bacteria data, we tried a machine learning approach. Before machine learning, bacterial abundances were logarithmically transformed. As bacterial data included 689 taxa at the genus level, such a large data volume would have tended to induce dimensionality for machine learning. Therefore, we first extracted feature-taxa by L1 regularized logistic regression (LASSO; least absolute shrinkage and selection operator) [22] as used previously for feature-taxa extraction [11]. We next identified IBS by random-forest analysis [23] using the extracted taxa. The random forest was packaged in a pipeline of Python scikit-learn to prevent data leakage [24] and subjected to repeated cross-validation (10-fold, one hundred repeats). A parameter of inverse of regularization strength for logistic regression was optimized by inner 5-fold cross-validation. The performance of the classifier was quantified by area under the receiver-operating characteristic (ROC) curves with an average of a thousand models. The source code for the prediction model is available from GitHub [25].

### 2.8. Statistical Analyses of the Fecal Microbiome to Determine the Featured Taxa in IBS Patients

To determine the featured taxa in IBS patients, we used the LASSO logistic regression algorithm as developed by Tap et al. [11]. This algorithm extracts features (bacterial OTUs) as non-zero coefficients from 100 LASSO models (trained in 10-fold cross-validation and ten repeats). As train and test data, our OTU-based data were filtered to remove OTUs that were detected in only one sample or less than 10 reads as a total amount for all samples. The labels for classification were IBS and healthy control. For comparison with the features of Swedish IBS patients, we extracted OTUs whose assigned taxonomy at the genus level had been commonly observed in the Swedish data [11] and our data. Each of the featured taxa (OTUs) was assessed by BLAST [26].

## 3. Results

### 3.1. Patient Characteristics and Clinical Status

Clinic and demographic characteristics for all of the subjects (85 IBS patients and 26 healthy controls) enrolled in this study are summarized in Table 1. Among the 85 IBS patients, 27 were diagnosed as IBS-C, 33 as IBS-D, 22 as IBS-M, and 3 as IBS-U according to the Rome III criteria. The various parameters including age, gender, and body mass index (BMI) did not differ significantly between the healthy controls and the IBS patients as a whole (Table 1).

The characteristics of the various IBS subtypes are also shown in Table 1. Age, gender, and BMI did not differ between IBS-D and IBS-M, but age was higher in IBS-C than in the healthy controls. Stool frequency was significantly lower in IBS-C than in controls, whereas it was significantly higher in IBS-D. Bristol Stool Scale score was significantly higher in IBS-D than in controls.

### 3.2. Biodiversity of IBS Subgroups and Healthy Controls

The data for microbiota diversity in fecal samples are shown in Figure 1. The Shannon index score was significantly lower in the IBS group (5.86 ± 0.65; 95% confidence interval (CI), 5.72–6.00) than in the healthy controls (6.15 ± 0.43; 95% CI, 5.97–6.32) (*p* < 0.05, *t* ~2.62; Figure 1A). The PD whole tree was also lower in the IBS group (31.3 ± 7.9; 95% CI, 29.6–33.0) than in the healthy controls (33.4 ± 4.2; 95% CI, 31.8–35.1), although not to a significant degree (*p* ~0.07, *t* ~1.82).

Comparison among the healthy controls and IBS subgroups showed that the microbial community in IBS-D patients was different from that in the controls (Figure 1B). Thus, the Shannon index and PD whole tree in the IBS-D group (Shannon, 5.62 ± 0.61; 95% CI, 5.49–5.83; PD whole tree, 29.5 ± 9.5; 95% CI, 26.1–32.9) were significantly lower than those in the controls (Shannon, 6.15 ± 0.43; 95% CI, 5.97–6.32; PD whole tree, 33.4 ± 4.2; 95% CI, 31.8–35.1) (*p* < 0.05, *t* ~3.97 and 2.15).

### 3.3. Short-Chain Fatty Acids in Feces Samples from IBS Patients

To identify potential biomarkers for IBS, short-chain fatty acids (SCFA) were analyzed in feces samples from healthy controls and IBS patients (Table 2). In the IBS group as a whole, the amount of propionic acid (C3) (11.6 ± 6.4; 95% CI, 6.8–15.1) and the difference between butyric acid and valerate (C4−C5) (5.1 ± 5.8; 95% CI, 1.3–7.2) were significantly increased compared to those in healthy controls (C3, 8.3 ± 4.4; 95% CI, 5.8–8.6; C4–C5, 1.4 ± 2.6; 95% CI, 0.1–1.9) (*p* < 0.05, *t* ~2.96 and 4.52), whereas acetic acid (C2) (36.9 ± 12.9; 95% CI, 26.3–43.9) was significantly reduced relative to that of the controls (42.0 ± 8.7; 95% CI, 36.7–45.7) (*p* < 0.05, *t* ~2.30).

When analyzed according to IBS subtype, the difference between butyric acid and valerate (C4−C5) was significantly increased in all IBS subtypes relative to that of the controls (*p* < 0.05). In addition, acetic acid in IBS-D patients (34.5 ± 13.0; 95% CI, 24.4–42.4) was significantly reduced relative to that of the controls (42.0 ± 8.7; 95% CI, 36.7–45.7) (*p* < 0.05, *t* ~2.64). In the IBS-M group, propionic acid was significantly increased (IBS-M, 12.9 ± 5.5; 95% CI, 8.5–17.3; Controls, 8.3 ± 4.4; 95% CI, 5.8–8.6) (*p* < 0.05, *t* ~3.07).

### 3.4. Distance of Microbial Composition between IBS and Healthy Controls

PCoA of unweighted UniFrac distances of microbial composition is shown in Figure 2A. The properties of healthy controls were positioned in the area where the level of PC1 and PC2 was less than 0.1 and 0.2, respectively. The properties of some IBS patients belonged to the same area, but those of others showed higher PC1 and/or PC2 levels, indicating that some IBS patients had healthy control-like properties whereas others were clearly distinguishable. Overall, IBS-C patients did not show high PC2 levels, but some showed high PC1 levels (>0.1).

Hierarchical clustering of unweighted UniFrac distance was also performed to visualize purely IBS clusters and IBS-healthy control mixed clusters (Figure 2B). The green clusters furthest apart from the IBS-healthy control mixed clusters were purely IBS clusters comprising 16 samples (8 D-type, 2 C-type, 5 M-type and 1 U-type) (Figure 2B).

The unweighted UniFrac distance between the healthy control group and the IBS group (0.74 ± 0.04; 95% CI, 0.740–0.743) differed significantly from the inner-distance of the healthy control (0.72 ± 0.03; 95% CI, 0.715–0.720) or IBS group (0.75 ± 0.04; 95% CI, 0.751–0.754) (*p* < 0.05, *t* ~14.5 and 10.7) (Figure 2C); thus, the distance within the IBS group was the longest and that within the healthy control group was the shortest. The distance of the IBS C-type from the healthy controls (0.73 ± 0.03; 95% CI, 0.732–0.736) was shorter than that of the other IBS types from the healthy controls (IBS-D, 0.75 ± 0.04; 95% CI, 0.748–0.753; IBS-M/U, 0.74 ± 0.04; 95% CI, 0.735–0.740) (*p* < 0.05, *t* ~9.18 and 1.99; Figure 2D).

### 3.5. Comparisons of Relative Abundance of Each Taxon between Healthy Controls and IBS Patients

With the univariate analysis, we found significant taxon at the genus level. The Welch’s test indicated statistical significance of relative abundances of several taxon existed between healthy controls and IBS patients (Table 3 and Figure 3).

### 3.6. Classification of IBS and Healthy Controls by Machine Learning with Featured Taxa and Short-Chain Fatty Acids

We attempted to establish a model for distinguishing IBS patients from healthy control groups using taxa-assigned and/or SCFA data (Figure 4). We first tested whether a combination of logistic regression and random forest would be better than either approach alone. We found that the combination of logistic regression and random forest for taxa-assigned data yielded an area under the curve (AUC) of 0.911 ± 0.088 (95% CI, 0.905–0.916), whereas the AUC obtained by logistic regression was 0.887 ± 0.112 (95% CI, 0.880–0.894) and that obtained by random forest was 0.846 ± 0.130 (95% CI, 0.837–0.854) (Appendix A). This confirmed that a combination of logistic regression and random forest was significantly better than either approach alone (*p* < 0.05, *t* ~5.20 for logistic regression and 12.9 for random forest); therefore, we decided to use this combination to establish a model for distinguishing IBS patients from control subjects.

The AUC obtained using only taxa-assigned data was 0.911 ± 0.088 (95% CI, 0.905–0.916) (Figure 4A). That obtained using both taxa-assigned and SCFA data was better at 0.920 ± 0.086 (95% CI, 0.915–0.926) (*p* <0.05, *t* ~2.47). Some SCFA levels in feces differed significantly between IBS patients and healthy controls (Table 2). However, the AUC obtained using only SCFA data was 0.733 ± 0.165 (95% CI, 0.722–0.743); subsequently, it was more difficult to distinguish IBS patients from healthy controls when using models with only taxa-assigned data (*p* < 0.05, *t* ~30.1).

We also attempted to distinguish healthy controls from each of the IBS subtypes using our machine-learning model with taxa-assigned data. IBS-D was well distinguished, and the AUC score was 0.980 ± 0.058 (95% CI, 0.976–0.983). IBS-C and IBS-M/U were also distinguishable with AUC scores of 0.854 ± 0.175 (95% CI, 0.843–0.865) and 0.906 ± 0.175 (95% CI, 0.895–0.917), respectively (Appendix A).

### 3.7. Comparison of Japanese IBS Featured Taxa with Swedish IBS

To determine whether the microbiomes in Japanese and Swedish IBS patients are similar or different, we extracted featured taxa using the LASSO logistic regression algorithm developed by Tap et al., which has been used to analyze Swedish IBS data [11]. The features extracted from our Japanese IBS data showed some bacteria that were not evident in the Swedish data, such as *Halomonas*, *Klebsiella*, *Dorea*, *Prevotella*, *Lachnobacterium*, *Ruminococcus*, *Collinsella*, *Streptococcus*, *Bifidobacterium,* and *Oscillospira* (Table 3 and Appendix A). Featured genera commonly observed in both the Swedish and our Japanese data were *Bacteroides*, *Faecalibacterium*, *Parabacteroides,* and *Blautia* (Appendix A).

## 4. Discussion

To establish an objective tool for diagnosis of IBS, we investigated the fecal gut microbiota profile in Japanese healthy subjects and IBS patients. Overall, the α-diversity of the gut microbiome was significantly decreased in Japanese IBS patients relative to that of healthy subjects and was lowest in IBS-D than in other types of IBS (Figure 1). However, the reduction of diversity was not as great as that in obesity [27] or patients with IBD [28], indicating that the dysbiosis in IBS may be comparatively subtle. Furthermore, since the gut microbiome data were based on analysis using 16S rRNA gene sequencing, exclusion of PCR bias may have been necessary. In a considerable proportion of IBS patients, the microbiome composition was similar to that in healthy subjects, although in some it was clearly different (Figure 2), similar to previous findings by Laubus et al. [29]. This variation may not be surprising, as a number of factors (e.g., race, diet, age, gender, social environment) that might possibly affect the gut microbiome profile play a role in the development of IBS, creating heterogeneity among patients [30]. The gut microbiota profile is known to be affected by race [31], and even within the same racial group, healthy individuals may show differences [12]. In this context, it was interesting to compare our data for Japanese IBS patients with those of Swedish IBS patients obtained using a similar study design [11]. This allowed us to extract some bacteria that were specific to Japanese IBS patients (Table 3 and Appendix A) and not observed in the Swedish study. Since the amplicon regions of the 16S rRNA gene differed between our study and the Swedish one, these two studies need to be compared with reference to the difference in the bioinformatics protocols employed. However, it was perhaps noteworthy that we detected a decrease of specific genera (*Bacteroides*, *Faecalibacterium*, *Parabacteroides,* and *Blautia*; Appendix A) that were common to both the Swedish and Japanese cohorts, suggesting that these genera may be highly reliable for distinguishing IBS patients from healthy controls.

There is also the issue of whether the difference in the gut microbiome profile is causative of IBS, or results from its development. This would appear difficult to address as both the gut microbiome profile and IBS pathophysiology are influenced by common environmental factors such as diet, psychological stress, lifestyle, and hormones [30]. In this context, fecal microbiota transplantation might seem to be an appealing approach for clarifying whether alteration of the gut microbiome is a possible cause of IBS. Interestingly, in germ-free animals, transplantation of the fecal microbiota from IBS patients has been shown to reproduce the visceral hypersensitivity or gastrointestinal dysmotility characteristic of IBS [32,33], indicating that the gut microbiome may indeed be a possible cause of IBS. However, Halkjær et al. have reported that transplantation of the fecal microbiota from healthy subjects to IBS patients conferred no benefit in terms of symptom relief [34]. Taken together, therefore, at least in humans, the existing data suggest that specific alteration of the gut microbiome profile may have no pathophysiological significance in IBS. On the other hand, among the environmental factors mentioned above, diet may have a critical impact on both the gut microbiome profile and IBS pathophysiology [35,36]. Using gnotobiotic methodology, Gordon’s group has suggested that diet plays an essential role in defining the gut microbiome profile [37], and moreover that certain dietary components such as fermentable oligosaccharides, disaccharides, monosaccharides, and polyols (FODMAP) not only change the composition of the human gut microbiome but also exacerbate the symptoms of IBS patients [38,39]. Unfortunately, the scope of the present study did not extend to analysis of the influence of diet on the gut microbiota profile in IBS patients, thus representing a qualitative limitation. However, not only diet but also various environmental factors influence the gut microbiome profile as well as IBS pathophysiology; therefore, it appears extremely difficult to clarify whether gut microbiome alterations are of crucial significance in this context.

The gut microbiota interacts with the host by producing SCFAs as mediators [40]. Indeed, it has been clarified that SCFAs act via specific receptors not only on epithelial cells but also immune cells in intestinal tissues [40,41], suggesting that SCFAs play a pivotal role in the pathophysiology of various gastrointestinal diseases. Our data indicated that propionic acid and the difference between butyric acid and valerate were significantly increased in IBS patients whereas acetic acid tended to be decreased (Table 2). When our patients were divided into groups according to IBS subtypes, those with IBS-D showed a significantly increased difference between butyric acid and valerate values, whereas acetic acid was decreased. As SCFAs are products of bacterial dietary fiber metabolism, their properties are determined by a combination of diet and gut microbiome composition. Although the existing data are conflicting, several studies have revealed that the propionic-acid-producing genus *Veillonella* is increased whereas the butyrate-producing Erysipelotrichaceae are decreased in feces from IBS patients [42,43,44]. In this context, the increase of propionic acid is consistent with previous reports [42,43,44], but we were unable to observe such alterations in the bacterial strains investigated here. Although a low FODMAP diet is useful for symptom relief in 50–80% of IBS patients [39], the mechanism of its effect is still unclear. Interestingly, a low FODMAP diet leads to a reduction of *Biffidobacteria* and butyrate-producing bacteria [45,46], which characterize IBS. Moreover, although a low FODMAP diet is likely to reduce the production of SCFAs, several studies have obtained conflicting results regarding the effects of such a diet [45,46,47]. Furthermore, we found that the differences in SCFAs among the various IBS subtypes were not so distinct (Table 2), implying that differences in fecal SCFA concentrations may not play a very significant role in determining the specific symptoms of IBS patients.

Our present goal was to establish a model for diagnosis of IBS using data for the fecal microbiome and SCFAs. Using LASSO regularized multiple logistic regression [22], we evaluated data obtained by 16 rRNA gene sequencing, and finally established a machine learning model for diagnosis of patients with IBS using fecal microbiome data (Figure 4B; sensitivity > 80% and specificity > 90%). We had initially expected that the SCFA data would have an additive effect on the diagnostic model, since SCFAs are also potential markers for IBS diagnosis [44]. However, as shown in Figure 4B, the SCFA data were of little additional advantage for our diagnostic model of IBS. This may not be surprising in view of the only slight differences in SCFAs between IBS patients and healthy subjects (Table 2). There is a need for powerful biomarkers that can aid in the objective diagnosis of IBS and/or prediction of the response to therapy, and numerous candidates (e.g., serum molecules, fecal metabolites, motility, psychological aspects) have been investigated [48,49,50]. Although data on fecal microbiota signatures have varied among studies of IBS patients, such differences may have been at least partly due to not only the design of such studies but also the geographic regions where they were conducted, as this aspect can affect diet and lifestyle [30]. Nevertheless, it is interesting that features observed at the phylum level, such as a *Firmicutes*/*Bacteroidetes* ratio, have been almost consistent among IBS patients [9], and some common findings at the genus level have also been reported for different cohorts such the Swedish and present Japanese ones. We have no exact explanation for why certain taxa are common to IBS patients in different geographic regions; however, a machine leaning system or statistical analysis may help to reveal the complex associations among IBS-related environmental factors and improve the sensitivity and specificity of tools for IBS diagnosis based on microbiome information.

In summary, we have clarified the gut microbiome characteristics of Japanese IBS patients and the SCFAs they produce. Moreover, we have established a machine learning model for diagnosis of IBS using fecal microbiome data. However, we concede that this study had several limitations. First, it lacked any functional investigations of the microbial community, and the number of control subjects was small, thus diminishing the study relevance. To advance this study, integration of meta-omics approaches such as metagenomics, metatranscriptomics, metaproteomics, or metabolomics would be required. Second, the lack of dietary information might have concealed any effect of diet on the gut microbiota profile. In addition, it might be questionable whether our diagnostic tool would be able to classify IBS patients into various subtypes. In this context, we aimed to create a model based on gut microbiome data and preliminary indications suggested that our strategy might also contribute to the establishment of a machine learning model for subclassification of IBS patients (Appendix A). Although further analyses will be needed before this diagnostic model can be established, our present work represents a first step towards devising an objective tool based on gut microbiome data for identifying IBS patients or individuals likely to develop the condition.

## Figures and Tables

**Figure 1 jcm-09-02403-f001:**
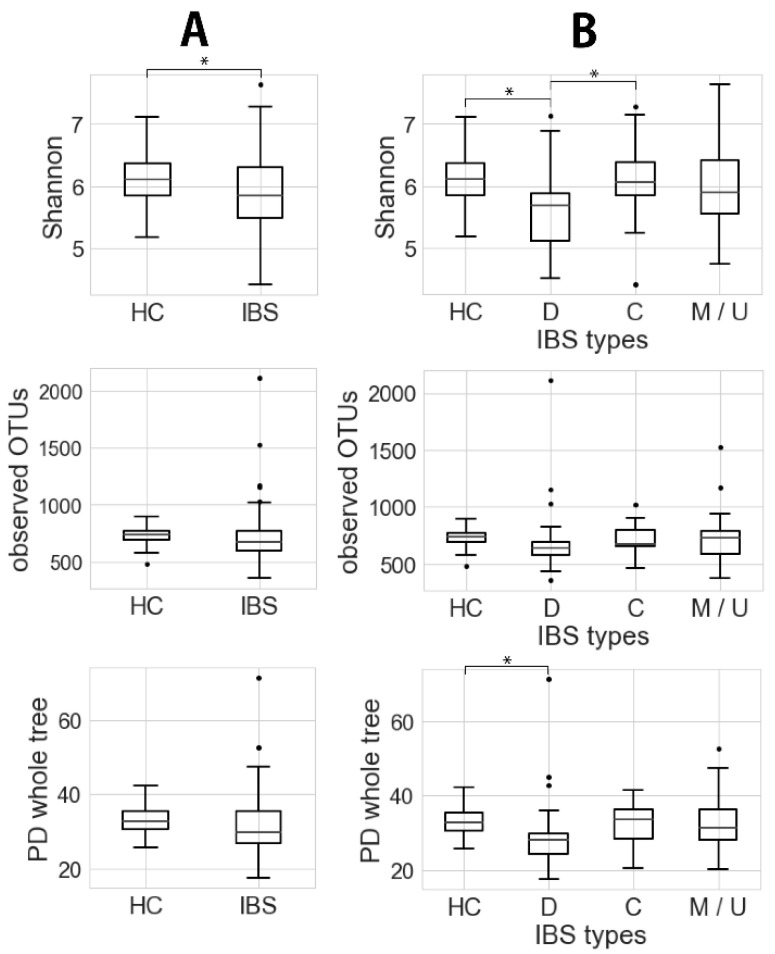
Microbiota α-diversity (Shannon index, observed operational taxonomic units (OTUs) and phylogenetic diversity (PD) whole tree) of (**A**) IBS (Welch’s test, * *p* < 0.05) and (**B**) IBS types (Welch’s test, * *p* < 0.05). HC, D, C, and M/U indicate healthy control, IBS-D, IBS-C, and mixture of IBS-M and IBS-U, respectively.

**Figure 2 jcm-09-02403-f002:**
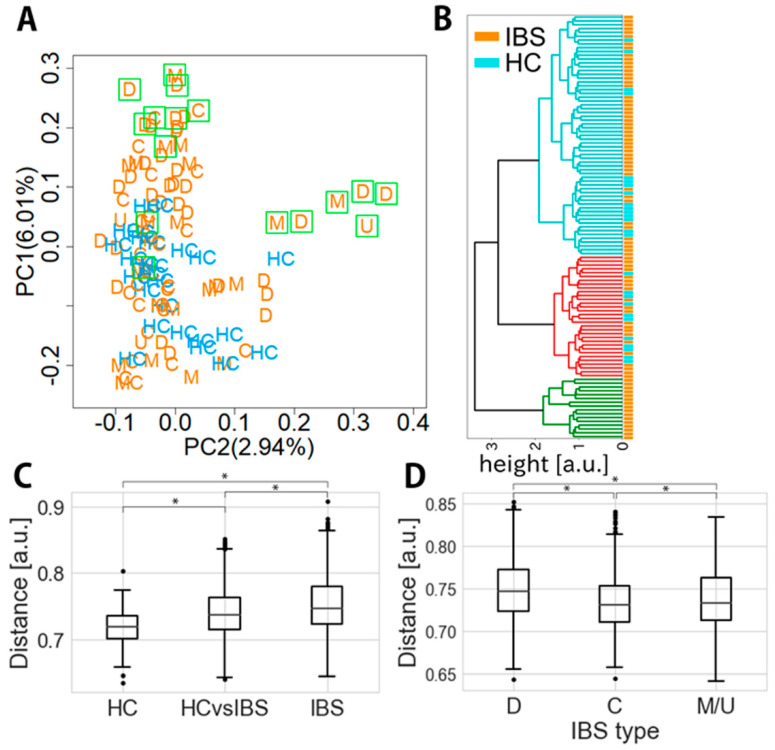
Distance of microbial composition between IBS patients and healthy controls. (**A**) Principal coordinate analysis (PCoA) of the unweighted UniFrac distance matrix from taxa-assigned data. Blue marks indicate healthy controls and orange marks indicate IBS types. Each orange letter indicates IBS subtype: D, IBS-D; C, IBS-C; M, IBS-M; and U, IBS-U. Green rectangles indicate the samples that belong to a green cluster coming from IBS samples alone in panel B. (**B**) Hierarchical clustering of unweighted UniFrac distance using Ward’s method for visualizing the relationship between IBS patients and healthy controls. A green cluster consisting of only IBS samples is shown as green rectangles in (**A**). (**C**) Unweighted UniFrac distance among healthy controls and IBS samples (Welch’s test, * *p* < 0.05). (**D**) Unweighted UniFrac distance of each IBS subtype from healthy controls (Welch’s test, * *p* < 0.05).

**Figure 3 jcm-09-02403-f003:**
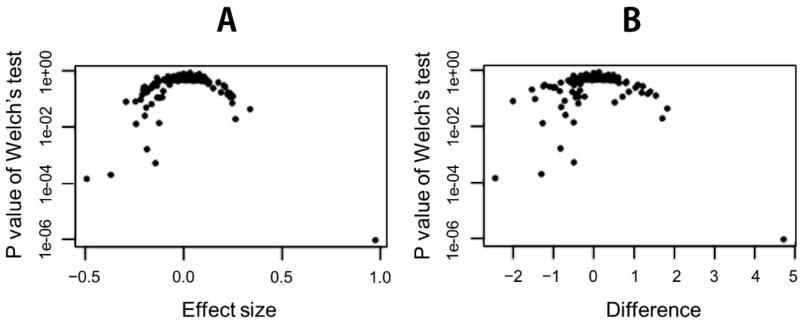
Effect plot examining univariate differences between IBS and healthy control groups. (**A**) The plot shows effect size versus the expected *p*-value of the Welch’s test. (**B**) The volcano plot shows the difference between groups versus the expected *p*-value of the Welch’s test.

**Figure 4 jcm-09-02403-f004:**
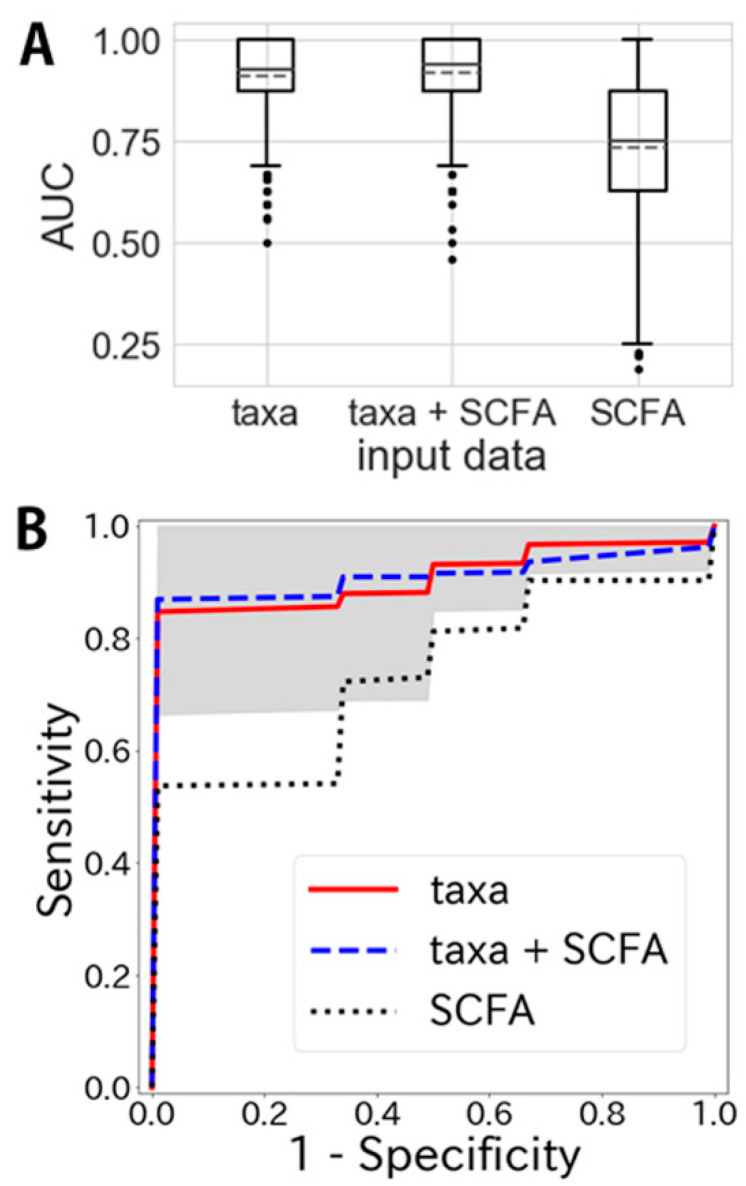
Area under the curve (AUC) scores and receiver-operating characteristic (ROC) curves for IBS prediction using taxa and short-chain fatty acid (SCFA) data. (**A**) Boxplots of AUC scores describing the prediction performance for IBS using taxa-assigned data, short-chain fatty acid data, and both. Solid lines in boxes indicate median and dashed lines indicate mean. (**B**) ROC curves describing specificity and sensitivity using taxa-assigned data, short-chain fatty acid data, and both. The gray shadow indicates the standard deviation of the ROC curve obtained using taxa-assigned data.

**Table 1 jcm-09-02403-t001:** Clinical and demographic characteristics for healthy subjects and irritable bowel syndrome (IBS) patients.

Factors	Healthy (*n* = 26)	IBS (*n* = 85)	*P* (*t*)	IBS-C (*n* = 27)	*P* (*t*)	IBS-D (*n* = 33)	*P* (*t*)	IBS-M (*n* = 22)	*P* (*t*)	IBS-U (*n* = 3)
Age	46.2 ± 10.6	51.3 ± 15.3	0.058 (1.93)	56.3 ± 15.2	0.007 (2.82)	49.7 ± 14.0	0.274 (1.10)	46.8 ± 16.7	0.873 (0.16)	56.7 ± 5.8
Sex (M/F)	9/17	37/48	0.562	8/19	0.925	17/16	0.301	11/11	0.433	1/2
BMI	22.2 ± 3.6	22.2 ± 4.2 (4 NA)	0.987 (0.02)	20.9 ± 3.5 (4 NA)	0.182 (1.35)	21.9 ± 3.2	0.742 (0.33)	24.6 ± 5.3	0.079 (1.81)	18.1 ± 1.1
IBS-symptom frequency (1/2/3)	NA	34/17/33 (1 NA)	NA	13/3/11	NA	12/7/13 (1 NA)	NA	9/7/6	NA	0/0/3
Stool frequency (stool per day)	1.48 ± 0.56	1.40 ± 0.91 (4 NA)	0.614 (0.51)	0.88 ± 0.72 (1 NA)	0.003 (3.19)	2.03 ± 0.82 (2 NA)	0.005 (2.92)	1.15 ± 0.80 (1 NA)	0.124 (1.58)	1.23 ± 0.46
Stool consistency (Bristol Stool Form)	4.15 ± 0.46	4.41 ± 1.61 (6 NA)	0.223 (1.23)	3.67 ± 1.88 (3 NA)	0.259 (1.16)	5.31 ± 0.71 (1 NA)	<0.001 (7.20)	3.95 ± 1.82 (2 NA)	0.630 (0.489)	4.00 ± 0.00

Data are shown as mean ± SD. The frequency of IBS symptoms was graded as 1, 3–9 days/month; 2, 10–19 days/month; 3, 20–every day/month. NA, not available. *t*, *t* value. BMI, body mass index. *p*-values for IBS-U were not indicated because of low numbers. IBS with constipation (IBS-C), IBS with diarrhea (IBS-D), mixed IBS (IBS-M), or unsubtyped IBS (IBS-U).

**Table 2 jcm-09-02403-t002:** Short-chain fatty acids in feces of healthy subjects and IBS patients.

Factors	Healthy (*n* = 26)	IBS (*n* = 81) ^†^	*P* (*t*)	IBS-C (*n* = 25) ^†^	*P* (*t*)	IBS-D (*n* = 33)	*P* (*t*)	IBS-M (*n* = 20) ^†^	*P* (*t*)	IBS-U (*n* = 3)
acetic acid	42.0 ± 8.7	36.9 ± 12.9	0.025 (2.30)	37.5 ± 11.2	0.114 (1.60)	34.5 ± 13.0	0.011 (2.64)	40.0 ± 14.2	0.516 (0.66)	39.8 ± 17.5
propionic acid	8.3 ± 4.4	11.6 ± 6.4	0.004 (2.96)	11.0 ± 6.6	0.096 (1.70)	10.8 ± 6.7	0.095 (1.70)	12.9 ± 5.5	0.004 (3.07)	17.1 ± 4.0
butyric acid	7.0 ± 3.4	6.5 ± 3.2	0.574 (0.57)	6.6 ± 3.1	0.668 (0.43)	5.6 ± 2.8	0.109 (1.63)	7.9 ± 3.9	0.402 (0.85)	7.3 ± 1.5
valerate	1.0 ± 0.9	1.1 ± 1.0	0.593 (0.54)	0.9 ± 0.8	0.797 (0.26)	1.1 ± 1.2	0.628 (0.49)	1.1 ± 0.9	0.726 (0.35)	2.4 ± 0.7
iso-butyric acid	0.7 ± 0.6	0.8 ± 0.5	0.302 (1.05)	0.9 ± 0.5	0.123 (1.57)	0.7 ± 0.5	0.970 (0.04)	0.8 ± 0.4	0.300 (1.05)	1.1 ± 0.8
iso-valerate	0.7 ± 0.5	0.7 ± 0.5	0.821 (0.23)	0.8 ± 0.6	0.456 (0.75)	0.6 ± 0.5	0.639 (0.47)	0.7 ± 0.4	0.873 (0.16)	0.9 ± 0.8
butyric acid- valerate	1.4 ± 2.6	5.1 ± 5.8	<0.001 (4.52)	4.4 ± 6.9	0.044 (2.10)	5.2 ± 5.7	0.001 (3.43)	5.0 ± 4.8	0.004 (3.10)	9.9 ± 4.3

Data are shown as mean ± SD. The *p* value for IBS-U was not indicated because of low numbers. *t*, *t* value. ^†^ Short-chain fatty acid data lacks 4 samples of IBS including 2 IBS-C and 2 IBS-M.

**Table 3 jcm-09-02403-t003:** Differences in abundance of single taxa between healthy controls and IBS patients.

Taxon genus level	Healthy Group (*n* = 26)	IBS Group (*n* = 85)	*P* (*t*)	IBS-C (*n* = 27)	*P* (*t*)	IBS-D (*n* = 33)	*P* (*t*)	IBS-M (*n* = 22)	*P* (*t*)	IBS-U (*n* = 3)
f_Halomonadaceae; g_Halomonas	0.00 ± 0.00	0.12 ± 0.18	<0.001 (15.38)	0.07 ± 0.10	<0.001 (5.53)	0.18 ± 0.24	<0.001 (12.86)	0.12 ± 0.13	<0.001 (9.33)	0.04 ± 0.04
f_Lachnospiraceae; g_Anaerostipes	0.41 ± 0.39	0.23 ± 0.45	<0.001 (5.67)	0.42 ± 0.70	0.008 (2.82)	0.08 ± 0.12	<0.001 (5.94)	0.21 ± 0.28	0.005 (3.07)	0.24 ± 0.17
f_Ruminococcaceae; g_Ruminococcus	4.41 ± 3.26	2.64 ± 2.93	<0.001 (3.99)	3.37 ± 2.99	0.120 (1.59)	1.72 ± 2.56	<0.001 (4.54)	2.94 ± 2.89	0.045 (2.08)	4.00 ± 5.05
f_Enterobacteriaceae; Other	0.02 ± 0.04	0.16 ± 0.57	0.001 (2.61)	0.12 ± 0.32	0.206 (1.28)	0.13 ± 0.18	0.002 (3.32)	0.28 ± 1.04	0.223 (1.23)	0.06 ± 0.07
f_Coriobacteriaceae; g_Collinsella	1.76 ± 1.36	1.23 ± 1.59	0.05 (2.95)	1.16 ± 1.50	0.022 (2.37)	1.01 ± 1.45	0.004 (3.00)	1.67 ± 1.95	0.304 (1.04)	0.98 ± 0.95

Data are shown as mean ± SD. The *p* value for IBS-U was not indicated because of low numbers. *t*, *t* value.

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
