# Peer review of "Usefulness of Machine Learning-Based Gut Microbiome Analysis for Identifying Patients with Irritable Bowels Syndrome"

_jcm, 2020, doi:10.3390/jcm9082403_

Round 1

Reviewer 1 Report

I appreciate the authors' effort in making the changes to the manuscript, making it more robust and conclusion is more appropriate to the data presented. I do not have any other comments and changes at this moment. Thank the authors for the good work.

---------------------

I have three main concerns (please see below) regarding the first draft of the manuscript, and I am pleased that the authors have addressed all these concerns in the revised version. At this point, I do not have any further comments to add, and I thank the authors for their effort in making the manuscript more robust and readable.

Original - My main concern with this study isn’t any functional investigation of the microbial community in this study, and as such associating the bacterial membership to IBS diagnosis is a huge assumption, since numerous studies has shown conflicting results. This is especially true since IBS is influences part by host genetics and part by environmental factors such as diet. As a result, bacterial diversity and taxonomy without any microbial function does not provide a lot of insights.  Therefore, authors will have to consider toning down their arguments in the manuscript. This is especially true since this OTUs are at 97% similarity.

The authors have highlighted the limitations of their analyses in the revised version. This is especially important in ensuring the readers get the accurate message.

Original - Another important issue is the lack of detailed values (mean/median, SD, significant test etc) in the Results Section. It is hard for readers to constantly going to and from the figures, supplementary data to figure out what is the mean/median and SD. Furthermore, the reporting of the test results is not clear. Overall, I think there is a lot of analysis lacking and tying everything together in the discussion.

In the revised version, the authors have added detail statistical values and describe the data in the table and figures. This will ensure readers find it easy to reference any analyses.

Original - The other important point is that the authors have not made their raw sequences, data publicly available. I think at the very least, this information should be in the supplementary section for the readers.

 I appreciate the authors making their data available on public database (NCBI SRA) to ensure reproducibility.

Author Response

Responses for the Comments from Reviewer 1:

Comments from Reviewer 1:

I appreciate the authors' effort in making the changes to the manuscript, making it more robust and conclusion is more appropriate to the data presented. I do not have any other comments and changes at this moment. Thank the authors for the good work.

Response to Reviewer 1:

     Thank you very much for your positive comments. Your suggestions were very precious for us to improve our manuscript. We would like to appreciate sincerely.

Reviewer 2 Report

The manuscript of Fukui et al. addresses a very interesting issue, i.e. the prediction of IBS on the bases of objective parameters, including the gut microbiota and metabolome, rather than subjective clinical symptoms.

In the study two independent cohorts of patients have been enrolled, namely a Japanese and a Swedish to account for differences in lifestyle and diet.

The manuscript is well written and the data are solid. Analysis are sound and conclusions are adequate to the data obtained

Major points:

_The analysis of the faecal metabolome are targeted and provide only a partial view of the complexity of the system.

It would be interesting to provide a more comprehensive metabolomic analysis or alternatively it could be sufficient to perform a PICRUSt analysis from the sequencing data.

Another possibility is to perform this prediction of the functional composition of the microbiota from the 16S sequencing data by using Piphillin.

Minor points:

_missing description of the Swedish cohort in material&methods

Author Response

Responses for the Comments from Reviewer 2:

We appreciate very much for your suggestions and believe that your constructive comments were very precious to improve and strengthen this study.

Major points:

Q1. The analysis of the faecal metabolome are targeted and provide only a partial view of the complexity of the system. It would be interesting to provide a more comprehensive metabolomic analysis or alternatively it could be sufficient to perform a PICRUSt analysis from the sequencing data. Another possibility is to perform this prediction of the functional composition of the microbiota from the 16S sequencing data by using Piphillin.

A1. Thank you for your professional comments. As you know, 16S rRNA gene sequencing simply reveals the type and its abundance of the bacteria strains contained in this samples. On the other hand, metagenome prediction using PICRUSt or Piphillin are able to predict the genes of gut flora, being useful to speculate the possible functions of those gut flora. In this context, integration of meta-omics approaches such as metagenomics, metatranscriptomics, metaproteomics, or metabolomics analyses will be better to predict the functions since these analyses evaluate the functional mediators and/or products from gut microbiome community. Of course, we know that these functional analyses would improve the quality of our study; however, it is currently impossible for us to do these functional analyses. In response to your comments, the following sentences were added in Discussion (5th paragraph) in the revised manuscript.        

“To advance this study, integration of meta-omics approaches such as metagenomics, metatranscriptomics, metaproteomics, or metabolomics would be required.”

Minor points::

Q1. missing description of the Swedish cohort in material & methods.

A1. We are sorry but we cannot find the description that you pointed out. The description is “To determine the featured taxa in IBS patients, we used the LASSO logistic regression algorithm developed by Tap et al. [11].” ? If so, of course, the algorithm is not perfectly same as the algorithm developed by Tap et al. Therefore, we changed the following sentence in Materials and Methods 2.8 in in the revised manuscript. 

From

“To determine the featured taxa in IBS patients, we used the LASSO logistic regression algorithm developed by Tap et al. [11].”

To

“To determine the featured taxa in IBS patients, we used the LASSO logistic regression algorithm as developed by Tap et al. [11].”

This manuscript is a resubmission of an earlier submission. The following is a list of the peer review reports and author responses from that submission.

Round 1

Reviewer 1 Report

This is a well-appreciated study that uses 16S rRNA amplicon and machine learning approach to associate fecal gut microbiome diversity and short-chain fatty acid (SCFA) with IBS patients. The author use machine learning to determine discriminative microbes between IBS patients and healthy control samples. The authors collected samples from 85 IBS patients and 26 healthy controls. The authors found that certain gut microbiome profile characterized Japanese IBS patients, while their approach showed no additional effects using SCFAs.

My main concern with this study isn’t any functional investigation of the microbial community in this study, and as such associating the bacterial membership to IBS diagnosis is a huge assumption, since numerous studies has shown conflicting results. This is especially true since IBS is influences part by host genetics and part by environmental factors such as diet. As a result, bacterial diversity and taxonomy without any microbial function does not provide a lot of insights.  Therefore, authors will have to consider toning down their arguments in the manuscript. This is especially true since this OTUs are at 97% similarity.

Another important issue is the lack of detailed values (mean/median, SD, significant test etc) in the Results Section. It is hard for readers to constantly going to and from the figures, supplementary data to figure out what is the mean/median and SD. Furthermore, the reporting of the test results is not clear. Overall, I think there is a lot of analysis lacking and tying everything together in the discussion.

The other important point is that the authors have not made their raw sequences, data publicly available. I think at the very least, this information should be in the supplementary section for the readers.

At this point I cannot support publication of this version of the manuscript. There are concerns that the presentation and stance the author has chosen to take will mislead readers who aren’t in the field as it does not accurately represent the result of the study. I hope they see the comments as an effort to help them improve the impact of their work.

Specific comments:

Methods section

Line 81: “were approved by the Ethical Review Board at our institutions.” Can the authors include the IRB number in this manuscript?

Line 93: “Fecal sampling, DNA extraction and sequencing” Can the authors make their raw sequences and data publicly available so that we can re-produce and validate the results?

Line 123: “P-values were calculated by the two-sided unpaired Wilcoxon rank sum test for testing group differences in diversity between the IBS patients and healthy controls.” Healthy control is 26 patients, while there are 85 IBS patients. With the pretty drastic unequal number of samples, Wilcox Rank test may not be the most effective to reveal the significance. Some tests are set up specifically to deal with the problem of unequal sample sizes and unequal variances:

- Dunnett’s T3 or Dunnett’s C can be used for pairwise comparisons. Use T3 for small samples, and C for larger samples.

- Games-Howell Pairwise Comparison Test: an extension of the Tukey-Kramer test to handle unequal variances. Although it has more power (narrower confidence intervals) than Dunnett’s tests, alpha inflation can be a problem.

- Tamhane’s T2: combines Sidak’s multiplicative inequality test with Welch’s approximate solution.

- Welch’s Test for Unequal Variances is a modified Student’s t-test. The modified degrees of freedom tends to increase the test power for samples with unequal variance.

Line 164: “from GitHub (https://github.com/Cykinso/paper supplements-ibs-classifier).” I checked the link, and it should be https://github.com/Cykinso/paper_supplements-ibs-classifier.

Results section

Line 191: “The data for microbiota diversity in fecal samples are shown in Figure 1.” Can the authors upload the number of reads per sample before and after the QC, as well as the final process? These values are essential in understanding if the microbial diversity is affected by the sequencing coverage per sample etc. Also, please have a supplementary table for all the relative abundance/ count values for each OTU and samples.”

Line 192 and throughout the Results section: “(P <0.05)” The results for Wilcox test should be (W = <value>, Z = <value>, p =<value>).  Can the authors please update accordingly? Also please report the results with mean/median with the SD, so that the authors have a sense of the data.

Line 226, Figure 2: “A) Principal coordinate analysis (PCoA) of the unweighted UniFrac distance matrix from taxa-assigned data. Blue marks indicate healthy controls and orange marks indicate IBS types.” What does those letters (IBS types) that are in squares mean?

Line 246: “the single taxon approach, the abundance of several fecal organisms” What is the reason for choosing these few genera? The authors can make it clearer to the readers by explaining their motivation.

Discussion section

Most of the discussion are repeats of what is in the results section, and the authors can either combine the results and discussion sections, or do more in-depth discussions to make the manuscript more robust. Also, it is hard to determine which part of the results/ figures the authors are talking about in the discussion section. It may be easier for the readers if the authors refer to the figures in the discussion section.

Line 299: “Overall, gut microbiome diversity was significantly decreased in Japanese IBS patients relative to healthy subjects.” This sentence/ conclusion is not absoultely true when just using 16S amplicon as an indication since there are sequencing, sample as well as PCR bias. The authors are advised their conclusion so that it is more clearly reflecting their data and not over-claim the conclusion.

Line 309: “In this context, it was interesting to compare our data for Japanese IBS patients with those of Swedish IBS patients obtained using a similar study design [11]. This allowed  us to extract some specifically featured bacteria in Japanese IBS patients (Table 3 and Supplementary Table 1) that were not observed in the Swedish study. On the other hand, we commonly detected a decrease of featured genera (Bacteroides, Faecalibacterium, Parabacteroides and Blautia; Supplementary Table 1) in both the Swedish and our Japanese cohorts, suggesting that these strains may be highly reliable for distinguishing IBS patients from healthy controls.” There is benefit for comparing with another study. However, the Swedish study uses V5-V6 region while the authors used V1-V2 in this study. I would be cautious in a direct comparison since different regions will yield different results. And there are influences from other factors such as bioinformatics protocols as well. The authors are advised to adjust their conclusion accordingly.

Line 328 and throughout the Discussion section: “This paragraph is redundant, as it is a repeat of methods and results on machine learning protocol. The authors could have either remove this, or intergrate the informaton in the methods or results section. The authors should discuss about the implications of the results they obtained based on the machine learning, for example why would there be certain taxa that may be related to the IBS patients etc.”

Line 335: “metagenomic data” The authors did not have any metagenomic data. These are marker gene (16S rRNA amplicon) data. Please change accordingly.

Reviewer 2 Report

The aim of this study was to establish an IBS machine learning prediction model based on 16S sequencing of faecal samples and SCFA analysis in 85 Rome III IBS and 26 healthy controls.  The authors report that the faecal microbiome alpha-diversity was significantly lower in the IBS group than the healthy controls. The SCFA analysis reports significant differences SCFAs in the IBS group when compared to the healthy controls. Bacterial features selected established a machine learning prediction model for identifying IBS patients from controls (sensitivity > 80%; specificity > 90%)

There are major limitations to this study:

  1. The n value for the controls is simply too low at n=26
  2. More importantly, while the inclusion/exclusion criteria are okay, Rome III was used whereas the entire field has moved to Rome IV. 
  3. The biggest limitation is what appears to be a failure to take dietary information into account. This is a serious limitation because of the impact of diet on the microbiome and on the symptoms experience by IBS patients. Without this information and analysis the study has limited value. 

Other points

Table 1 is not very informative and should be presented more clearly based on the actual Bristol Stool Scale score and frequency

Table 2 The IBS n value is 81 not 85. Explain.

There is further confusion in Fig 2/Table 3 where the healthy group appears to be n=28 when the cohort is supposed to be n=26 and the IBS group is 83 not 85

Re SCFAs - Once again, a major failing is the lack of dietary data and analysis which obviously influence the microbiota and SCFAs.

In line 335 they state ‘In the present study, we evaluated metagenomic data using LASSO regularized multiple logistic’ but it was 16S data not shotgun.
